

# The impact of the planetary β-effect on the tilting vertical structure of a mesoscale eddy

Shengmu Yang[1,2], Jiuxing Xing[1], Shengli Chen[1], Tian Jiwei[3], and Daoyi Chen[1,2]

[1]Shenzhen Key Laboratory for Coastal Ocean Dynamic and Environment, Graduate School at Shenzhen, Tsinghua University, Shenzhen 518055, China
[2]School of Environmental Science and Engineering, Tsinghua University, Beijing 100084, China
[3]Physical Oceanography Laboratory/CIMST, Ocean University of China, Qingdao 266100, China

*Correspondence to*: Daoyi Chen (chen.daoyi@sz.tsinghua.edu.cn)

**Abstract.** Tilting mesoscale eddies in the South China Sea have been reported recently from observed field data. The mechanism of the dynamic process of the tilt, however, is not well understood. In this study, the influence of planetary β on the vertical structure of mesoscale eddies and its mechanism is investigated using theoretical analysis and numerical model experiments based on the MIT General Circulation Model (MITgcm). The results of the both approaches show that vertical motion due to the planetary β effect and nonlinear dynamics causes a pressure anomaly in the horizontal domain which triggers the tilt of the eddy axis. The tilting distance extends to be the radius of the eddy maximum velocity. In addition, the vertical stratification is another key factor in controlling the tilt of a mesoscale eddy. External forcings such as wind and inflow current are not considered in this study, and topography is included only in a realistic South China Sea model. Therefore, mesoscale eddies with large vertical depth should have the similar axis tilt character in open oceans under the β-effect.

## 1 Introduction

Oceanic mesoscale eddies are the most energetic form of circulations which are ubiquitously observed in the ocean (Zhang et al., 2016; Chelton et al., 2011). In the South China Sea (SCS), especially to the southwest of Taiwan, warm mesoscale baroclinic anticyclonic eddies are formed frequently. After its formation, an eddy propagates to the southwest along a bathymetry contour of the continental shelf. Recent results from the observed field data show that the vertical axis of an eddy tilts to the south-west from the sea surface to the bottom (Zhang et al., 2016). Lin et al. (2015) studied three-dimensional properties of mesoscale eddies in the South China Sea and the cross section of a warm eddy's vertical structure also showed the clear axis tilt although they did not discuss the issue. Elsewhere, Savchenko el al. (1978) found the spatial structure of a cyclonic eddy with tilted axis derived from direct instrumental observations in the Antarctic Circumpolar Current. Therefore, the vertical eddy axis tilt may be a common phenomenon for mesoscale eddies, but its mechanism is not clear yet.

Research on mesoscale eddies in the ocean has been carried out for decades. Great progress has been made from analyses of observations and from theory (Chaigneau et al., 2011; Nan et al., 2011; Zhai et al., 2010; Casella et al., 2011; Wu and Chiang,





2007). Mesoscale eddies propagate westwards under the influence of the β-effect (Sutyrin et al., 2003; Chelton et al., 2011). At the same time, a nonlinear meridional component of motion arises out of initial dispersion tendencies in the eddy, causing asymmetry of the eddy. The nonlinear momentum advection acts meridionally in the β plane (northward for cyclones, southward for anticyclones; Smith and O'Brien, 1983; Cushman-Roisin et al., 1990). As a result, a warm eddy moves to the

5 southwest while a cold eddy moves to the northwest (Wei and Wang, 2009; Yang et al., 2017). At the same time, the horizontal shape of the eddy becomes asymmetrical with respect to the longitude direction (Nof, 2010). When an eddy moves on the β plane, its maximum speed of propagation is limited to $-\beta R_d{}^2$ ($R_d$ is the Rossby radius of deformation; Nof, 2010; Smith and O'Brien, 1983). Characteristics of eddy motion in open oceans have been studied previously, but there is little research on the influence of planetary β on the vertical structure of the eddy.

In addition to the planetary β-effect, the motion of an mesoscale eddy is also affected by the topography (Oey and Zhang, 2004). The eddy's trajectory and structure may change due to the interaction with a continental slope, an island or a seamount. The interaction between an eddy and topography has been investigated by many researchers based on satellite observations (Dewar, 2002; Itoh et al., 2011), laboratory experiments (Cenedese, 2002) and numerical models ( Sutyrin and Grimshaw, 2010; Herbette et al., 2005; Luo and Liu, 2006; Hyun and Hogan, 2008; Jacob et al., 2010). Because the eddy propagation near

shelf follows the regional sloping bottom topography, the topographic β effect is an important local factor on controlling eddy's movement in coastal region.

With the help of satellite observations, the character of mesoscale eddy motion has been extensively studied. By contrast, because of the scarcity of the observational data for the full depth structure of an eddy, there is little research focused on the eddy vertical structure evolution.

Motivated by the observation of the eddy's tilting axis in the South China Sea (Zhang et al., 2016), and at the same time considering that the β-effect is more general than that of topography in influencing eddy structure, our main attention will focus on the influence of planetary β on eddy vertical structure evolution. Topography is included only in a realistic South China Sea model.

The paper is organized as follows: A simple theoretical analysis for the mesoscale eddy axis tilt is presented in Section 2.

Section 3 and section 4 provide a description of the numerical model we used and results from the model, respectively. On the basis of the theory and model results, Section 5 applies the model to the South China Sea with real topography conditions. Section 6 contains the conclusions and discussion.

## 2 An argument based on a simple theory

For a baroclinic mesoscale eddy, the first order force balance is between the horizontal pressure gradient and the Coriolis force,

i.e., the geostrophic balance, namely:

$$-fv = -\frac{1}{\rho_0}\frac{\partial p}{\partial x} \qquad (1)$$





$$fu = -\frac{1}{\rho_0}\frac{\partial p}{\partial y} \qquad (2)$$

where u, v are the zonal and meridional velocity respectively, p the pressure, f the Coriolis parameter, $\rho_0$ the reference density.

Using the continuity equation, the vertical velocity can be estimated from

$$\frac{\partial w}{\partial z} = -\left(\frac{\partial u}{\partial x} + \frac{\partial v}{\partial y}\right) \qquad (3).$$

On an f-plane, with assumption of the eddy being in geostrophic balance and the vertical velocity at the sea surface being zero, the vertical velocity is zero at all depths as f is a constant in the equations (1) and (2). In a more realistic situation, the Coriolis parameter increases from the equator to the pole. On the β-plane, we assume that f is a linear function of distance in the longitude direction, namely, $f = f_0 + \beta y$.

Therefore, the vertical velocity is given by $\frac{\partial w}{\partial z} = -\left(\frac{\partial u}{\partial x} + \frac{\partial v}{\partial y}\right) = \frac{\beta}{f}v$ .

The variation of the Coriolis parameter with latitude allows a vertical velocity gradient to exist in the geostrophic current in the ocean; the vertical velocity gradient is proportional to the meridional velocity and to β.

Take an anticyclonic warm eddy as an example. The β-effect can cause horizontal convergence and downward vertical motion at the western part of the eddy, while there is horizontal divergence and upward vertical motion at the eastern part. Hence a mesoscale eddy is in ageostrophic balance on the β plane, the time variation of the flow is dictated by much smaller ageostrophic components to the flow, including the vertical motion (Martin and Richards, 2001). On the other hand, the mesoscale eddy radiates energy in the form of Rossby waves while propagating southwestwards. As the longer wavelength Rossby components of the eddy travel faster, there is a weakening of pressure gradient on the leading western side of the eddy and steepening of pressure gradient on the trailing eastern side (Smith and O'Brien, 1983). The asymmetry in the pressure field and Radiation Rossby waves favor the convergence and downward vertical motion on the northern side of the eddy while divergence and upward vertical motion on the southern side (Fig. 1).

Combining the β-effect on the meritional flow and radiation Rossby waves effect, the distribution of horizontal convergence and downward motion is in the northwest of the eddy and horizontal divergence and upward motion is in the southeast. The schematic diagram of Fig. 2 shows the evolution process of the warm anticyclonic eddy. The ageostrophic velocities, in turn, are driven by two mechanisms: the non-linear momentum advection and the β-effect. Because of the existence of the thermocline, the vertical stratification is nonlinear through depth. The vertical motion induces lifting/lowering of isopycnals. As a result, compared with the eddy on a f-plane, in the upper layer of the eddy, the local horizontal convergence and divergence causes the northward velocity of the eddy to strengthen and the southward velocity to weaken before the eddy tilt is in a new balance. And in the subsequent adjustment process, the southward velocity could be balanced by larger pressure gradient on east side. In the lower layer, a local relative high pressure anomaly is caused by the downward vertical motion in the northwest of the eddy with lowering of isopycnals, and the upward motion causes a local relative low pressure anomaly in the southeast with lifting of isopycnals. In consequence, the northward velocity weakens and southward velocity strengthens.



Because of the variation of horizontal velocity in the eddy, there is overall more northward momentum than southward momentum in the upper layer and less in the lower layer. The integral of the Coriolis force on horizontal layer of the eddy is towards east at the upper layer and west at the lower layer. As a result, the upper layer shifts to the east and the lower layer shifts to the west. For a baroclinic mesoscale eddy, a sketch of the changes of the vertical structure on a latitudinal cross section is shown in Fig. 3. In a holistic view, the vertical structure of the eddy tilts westwards with depth. At the same time, the magnitude of the eddy tilt is constrained by the baroclinic pressure which is caused by the eddy tilting.

## 3 Numerical model experiments

### 3.1 The eddy structure

As mentioned before, this research is inspired by the observation of a tilting eddy in the SCS. Much previous research has explored the mesoscale eddy structure in the SCS, especially its three-dimensional density distribution (Hwang and Chen, 2000; Wang et al., 2005). Using the observational data as a guide, following Yang et al. (2017), we construct an idealized mesoscale warm eddy as the initial condition for the model: an axisymmetric Gaussian-type profile for the temperature field, namely,

$$T(z) = T_b(z) + a_z e^{-\frac{x^2+y^2}{2L^2}} \qquad (4)$$

where $T_b(z)$ is background temperature; $a_z$ is a function parameter varying with depth (z) and L is a constant, $6.0 \times 10^4$ m; x, y and z are position coordinates.

The eddy's initial tangential velocity is calculated using the thermal wind balance with a zero velocity at the ocean bottom. Figure 4 shows the temperature and azimuthal velocity distribution on the cross section through the eddy centre. The initial eddy is 120 km in diameter and 1000 m in depth with a total water depth of 2000 m, with the maximum surface velocity being about 1.1 m/s, and the maximum surface elevation being 0.4m.

### 3.2 Numerical model and initialization

In this study, the MIT General Circulation Model (MITgcm) was used to examine the vertical structure evolution of mesoscale eddies under the influence of the β-effect. For mesoscale dynamic processes, the non-hydrostatic dynamics play minor roles, hence the hydrostatic form of the model was used in our problem. The model domain was $500 \times 450$ km$^2$ with 2000 m depth. The horizontal resolution was 2.5 km; in the vertical dimension, 28 levels were set with a 50 m resolution in the upper 1000 m and gradually became coarser closer to the bottom layer. The model boundaries all were open, and the Orlanski radiation condition was used. The Coriolis parameter f=$5 \times 10^{-5}$ s$^{-1}$ and planetary β=$2.0 \times 10^{-11}$ m$^{-1}$s$^{-1}$; β was set to zero while on the f plane. In the horizontal, we used Smagorinsky viscosity with a parameter of 0.2 (Yang et al., 2017). The Laplacian diffusion of heat was used with a constant coefficient of $10^{-4}$ m$^2$/s. In the vertical, the eddy viscosity was $5.0 \times 10^{-4}$ m$^2$/s. For the temperature



equation, the vertical eddy diffusivity was $10^{-4}$ m$^2$/s and horizontal eddy diffusivity was set to zero. A flat bottom was used in the study of the β-effect, and at the end, we apply our model on a realistic topography on the β plane.

## 4 Results from the flat bottom experiments

In the study, we set up two sets of experiments: one on the f plane and the other on the β plane. Both experiments are with a flat bottom (here, we do not take topography effects into account). The first set of the experiments is a baseline and the second set of experiments was carried out to test the influence of β-effect on the vertical structure of the eddy. The two sets of the experiments have the same initial conditions; the eddy centre is located at (x=332 km, y=320 km) at the beginning of the model integration. We run the model for 50 days, although after 10 days the eddy vertical structure is well established. In this section, therefore, the model results of day 10 are presented.

### 4.1 The f plane

On the f plane, the eddy remains stable without external forcing after adjustment at the beginning of the model integration. It stays in the original position and does not move in 50 days; the velocity decreases a little because of the model viscosity. A cyclonic eddy is induced at the lower-layer by the geostrophic adjustment in response to pressure anomalies caused by the upper-layer mass loading. Figure 5 shows the distribution of horizontal velocity across the eddy centre, depicting two eddies: one is anticyclonic in upper waters and the other is cyclonic at greater depth. The anticyclonic eddy is mainly above 1000 m depth, and the cyclonic eddy is below 1500 m depth.

In the horizontal, the velocity distribution is symmetrical about the eddy centre. In the vertical, the axis of the eddy centre, which is defined as the central point of zero velocity at each layer, is erect as can be confirmed in Fig. 6. The v-velocity distribution (positive is northward, and negative is southward) along x-axis through the eddy centre at different depths intersect at one point which is the eddy centre. It means the centre of the eddy at the different depths all are vertically aligned. Therefore, the vertical structure of the eddy keeps upright on f-plane and the eddy is said to have a 'vertical' axis.

Figure 6 shows that in the eddy core region (where the circulation speed increases along the radius, 300 km<x<360 km), the velocity increases linearly with horizontal distance from the centre and then decreases exponentially in an outer annulus. The velocity has maxima at x=300 km and 360 km. For the upper anticyclone, the maximum velocity of the eddy at 100m depth is about 1.7 m/s and decreases with depth. For the lower layer, below the depth of 1200 m, the water moves cyclonically. In the bottom cyclone, the horizontal velocity distribution at different depths is basically the same. The strength of the lower-layer cyclone is decided by the initial conditions of the upper-layer anticyclone including its horizontal scale and depth. In the model, the maximum velocity of the cyclone is approximately 0.2 m/s.

Because the eddy has the same scale at different depths, it is a cylindrical vortex. According to the theoretical analysis and model results, the eddy will not move and the flow in the eddy is stable on the f-plane. The vertical structure of the eddy remains upright. In a more realistic ocean, a mesoscale eddy propagates westward at the Rossby wave speed. In this way, the



β-effect is an important factor in the movement of an eddy and its internal dynamic process. The influences of planetary β on the eddy structure are discussed in next section.

## 4.2 The β plane

In the second set of numerical experiments, a warm eddy with the same initial conditions as the first experiment on a flat
bottom was tested on the β-plane (β=$2.0\times10^{-11}$ m$^{-1}$s$^{-1}$). When an isolated eddy propagates in an open ocean with a flat bottom, it is well known that the movement is governed by the planetary β and nonlinear effects (e.g. Chang et al., 2012; Hyun and Hogan, 2008). From the results of the model, the warm eddy generally moves to the southwest which agrees with previous studies. Therefore at a different latitude which with different planetary β value, the eddy has different maximum propagation speed. At the same time, the eddy leaves a trail of Rossby waves in its wake which means the mesoscale eddy radiates energy
in the form of Rossby waves (Flierl, 1987; Smith and O'Brien, 1983; Mcwilliams and Flierl, 1979).

For the initial condition, the velocity value is symmetrical about eddy centre which is the same as on the f-plane. From the theoretical analysis, the β-effect could induce vertical motion in the eddy. The model results for the vertical velocity distribution in the eddy at 500 m depth are shown in Fig. 7. The horizontal distribution of vertical velocity caused by the β-effect is not symmetric in longitude, but shifts clockwise in azimuth. As a result, negative vertical velocity is located in the
northwest of the eddy and positive vertical velocity in the southeast which corresponds with our theoretical analysis.

Assuming the vertical velocity is zero at the surface, then it increases gradually with water depth. According to the model results, the maximum value of the vertical velocity at 100 m is $3.5\times10^{-4}$ m/s while at 1000 m it is $6.8\times10^{-4}$ m/s, which is about twice the velocity at the subsurface.

From the v-velocity distribution along x-axis through the centre of the eddy (Fig. 8), we can see that the velocity decreases
with depth and horizontally, and is asymmetrical about the eddy centre, very different from the velocity distribution on the f plane. Near the surface (100 m depth), the maximum velocity on the left side (west side) of the eddy is 1.4 m/s but 1.37 m/s on the right side (east side). The difference between left and right sides' maximum velocities increases with depth. In the lower layer (1000 m) the maximum velocity on the right is 0.26 m/s which is greater than on the left side (0.17 m/s). In adition, the v-velocity distribution at different depths indicates that the eddy centre (the point that the velocity is zero) shifts to the left
(west) along the depth. At 1000 m, the eddy centre shifts 30 km relative to the eddy centre at the subsurface. The cyclone induced by the upper anticyclone is also shown in Fig. 8 with the red lines showing the velocity distribution of the cyclone along the x-axis. The maximum velocity in the cyclone is 0.2 m/s . The centre of the cyclone is shifted to the right by 30km which is similar to the amount of the anticyclone tilt in the upper layer. The centre of cyclone (the position where the circulation speed is zero) is located below the position of the maximum velocity of the right side of the anticyclone.

The axis tilt is also shown in the vorticity variation in the upper and the lower layer (Fig. 9). The vorticity is positive in the inner eddy and negative in the outer eddy. At the sea surface, the centre of the eddy is located at (x=325 km, y=302.5 km) and is shifted to the west at 1000m depth (x=305 km, y=292 km). In addition, because of the tilt of the eddy and the possible



baroclinic instability, the lower parts of the anticyclone decay through advection and dispersion while the upper parts remain circular.

According to the model results, the horizontal scale of the eddy is 150 km (here we use temperature anomaly to define the eddy scale) and the maximum velocity diameter is 70 km. The horizontal distance of eddy axis tilt is about the maximum velocity radius of the eddy which is about 35 km. The east-west cross section of v-velocity distribution through the centre of the eddy is shown in Fig. 10. The vertical structure of the eddy presents a clear tilt to the west with depth.

The axis of the anticyclonic eddy remains upright in the surface layer, and is displaced to the southwest from 600 m depth to the depth of 1200 m, which is below the thermocline. For the cyclonic eddy in the deep-layer, the eddy centre drifts to the east relative to the upper anticyclone and the initial vertical structure of the eddy.

Figure 11 (a) shows the vertical profile of temperature along the centre of the eddy and background field. The temperature changes rapidly from the sea surface to the depth of 475 m and there is a weakening temperature gradient in depths of 500 m to 1000 m. The temperature is nearly constant below 1000 m. Correspondingly, the vertical velocity is near constant in the upper 125-475 m and increases with the depth in 500 to 1000 m (Fig. 11 (b)). The eddy axis displacement does not increase linearly with depth (Fig. 11 (c)). In the upper 450m, although the vertical axis is disturbed, it does not tilt. Below 500 m, the vertical axis begins to move westwards. This suggests that the axis tilt mainly occurs below the thermocline (because the salinity in the model is constant, the position of the thermocline and the pycnocline are the same). By comparing Fig. 11 (b) and (c), the maximum vertical velocity and the axis displacement have similar variation with depth. Correlating the vertical velocity in the eddy and the axis displacement, the coefficient of determination is 0.91. It indicates that in this case (without topography) the vertical velocity caused by the β-effect and thermocline position are significant for the vertical axis tilt of the eddy.

In an extra model experiment, we used a different vertical temperature profile, namely a linear vertical temperature profile without a strong thermocline. The v-velocity distributions along the x-axis through the centre of the eddy at different depths are shown in Fig. 12. For the upper anticyclone, at the same depth, the maximum velocity on the right (east side) is always larger than on the left (west side) from the surface to bottom which is different from the previous case. Because the maximum velocity difference between left and right sides increases with water depth, for the offset distance of the eddy axis, the lower part is larger than the upper. Therefore, the vertical structure of the eddy also shows a weak oblique tendency. The distance of the eddy tilt is about 10 km. By comparing the eddy vertical structures with the linear and nonlinear stratification, it is evident that the ocean stratification is an important factor in controlling the vertical structure tilt of a mesoscale eddy.

## 5 Application to the SCS: eddy trajectory and 3D structure in the SCS

The kinetic characteristics of eddies originating from southwest of Taiwan have been widely studied (Wu and Chiang, 2007; Zhang et al., 2016). Generally, warm eddies propagate southwestwards along the bathymetry contours in the South China Sea, with average speed about 0.064 m/s (Nan et al., 2011).



Here, we set up a numerical model using the real topography of the South China Sea on the β-plane. The maximum model water depth used is 2000 m. As previously, β is $2.0 \times 10^{-11}$ m$^{-1}$s$^{-1}$ and f=$5.0 \times 10^{-5}$ s$^{-1}$. A warm eddy starts southwest of Taiwan with a diameter of 120 km (Fig. 13). The vertical depth of the eddy is about 1000m and the model runs for 50 days with the other model parameters the same as in previous runs with nonlinear stratification.

The results of the model with real topography (Fig. 13) show that the eddy moves towards the southwest along the continental slope. The path of the eddy is consistent with previous research results. The average propagation speed of the eddy during the 50 days is about 0.057 m/s. Under more realistic (natural) oceanic conditions, the eddy's motion is affected by surface wind stress and background advection in addition to the topography and the β-effect. Our results (with 0.057 m/s being a close match to the observed 0.064 m/s) indicate that, for the general situation, the β-effect and topography are the main regular factors in

controlling the eddy movement rather than other external forcings in open oceans.

In addition, recent observational data shows that the full structure of the warm eddy does not remain vertical during its evolution. The vertical axis of the eddy tilts to the southwest with depth (Zhang et al., 2016). Motivated by this new finding we investigate the details of the vertical structure of the eddy in the South China Sea during its evolution in the model (Fig. 14). At t=10 days, the centre of the eddy is located at (x=318 km, y=308 km) at the depth of 100m and at (x=262 km, y=312

15   km) at a depth of 1000m (Fig. 14 (a)(c)). The eddy axis tilts to the southwest from the sea surface to bottom. From the sea surface to 1000 m depth, the tilting distance reaches ~60km. Because the lower part of the eddy structure is influenced by the deep circulation and the bottom topography, the eddy tilting distance is larger than that on a flat bottom (~30km). It is interesting to note that in the lower layer (Fig. 14 (c)), there is a cyclonic eddy (roughly equivalent strengths) generated at the east side of the anticyclone. It seems that the pressure anomaly caused by tilting of the eddy is more likely to cause energy

radiation at the lower part of the eddy. At t=50 days, the eddy propagates southwestwards to the position (y=208 km, y=173 km) at a depth of 100m and to (x=155 km, y=170 km) at a depth of 1000m (Fig. 14 (b) and (d)). The eddy still keeps tilting westwards from the sea surface to the bottom with a relative displacement of ~60 km. At t=50 days, there is a Rossby cyclonic eddy formation at the eastside of the eddy at full depth. And in the lower layer, a large scale cyclonic circulation is formed at the east side of the anticyclonic eddy (Fig. 14 (d)).

**6 Summary and discussion**

Motivated by the recent field observations of the eddy tilting structure in the South China Sea, a simple theoretical analysis and numerical experiments are used to investigate the mechanism of the tilt on the β-plane. Warm eddies are common in the area to the southwest of Taiwan and are often shed from Kuroshio meanders. Initially a flat bottom and an idealized mesoscale eddy balanced with the temperature structure are used in the numerical model (MITgcm). The model results correspond well

with the theoretical analysis and both show the eddy vertical structure tilts westwards. The distance of the eddy tilt is ~35 km which approximates the maximum velocity radius.



The mechanism of the eddy tilt is that the vertical motion caused by the nonlinear advection of momentum and the β-effect induces a pressure anomaly which can adjust the ageostrophic velocities of the eddy. As a result, the integral of Coriolis force is towards the east for the surface layer and towards the west for bottom layer water. On the overall view of the vertical structure of the eddy, the vertical axis tilts westwards with depth for anticyclonic eddies.

The distribution of the tilt displacement with depth shows the eddy tilt mainly occurs below the thermocline. At the same time, a high correlation is also found between the tilt displacement and vertical velocity in the eddy, indicating that the vertical velocity is an important factor in eddy tilt.

In addition to the β effect, the topography is another important factor causing eddies' vertical structure evolution. The tilting distance in the model with a realistic northern slope of the SCS is almost twice that with a flat bottom. The eddy-slope
interaction generates a deep cyclone and also affects the eddy motion and its structure, making the dynamics more complex. Therefore the effect of bottom topography on the vertical structure of an eddy merits further investigation.

## Acknowledgements

This work was supported by the National Key Basic Research Program of China (Program 973) (grant 2014CB745001), the Environmental Protection Special Funds for Public Welfare (201309006), and Shenzhen Government (201411201645511650;
2016M591159; KQJSCX20170720174016789; CXZZ20140521161827690). Authors would like to thank Prof. J Huthnance of NOC (UK) for the insightful comments and significant improvement on English writing.

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

**Figures:**




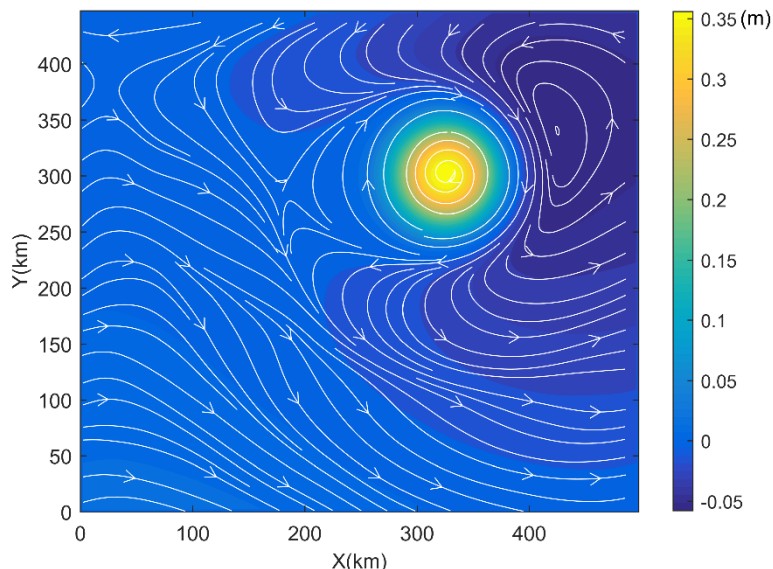

**Figure 1. A snapshot of sea level (in colors) and streamline at the sea surface while a mesoscale eddy is propagating to west (modelled by the MITgcm. The numerical model is described in section 3).**

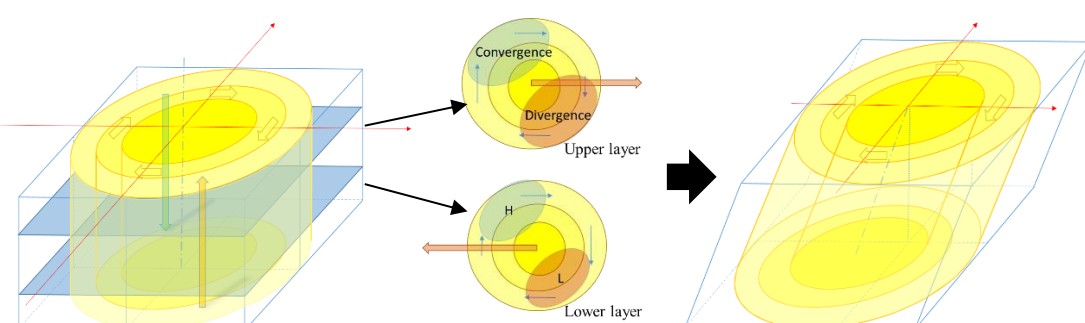

**Figure 2. The schematic diagram of vertical structure evolution of a warm eddy (H, L the local high and low pressure anomaly).**





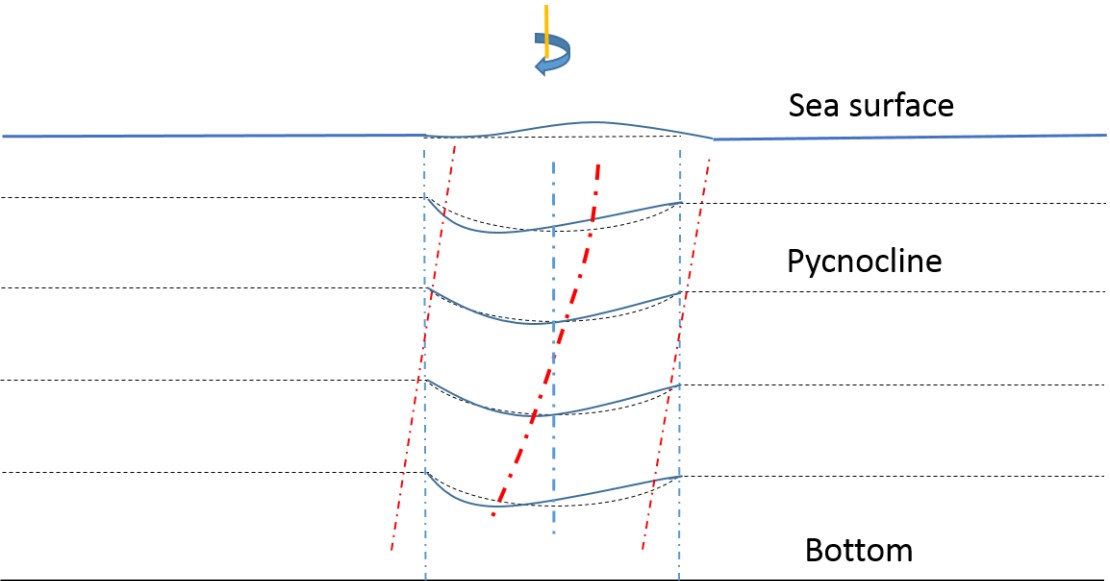

**Figure 3. A sketch of the eddy structure variation from f plane to β plane. The dotted lines show the original density distribution on the f plane and solid lines present the density distribution on the β plane. The red dot lines show the changing trend of the eddy's vertical structure.**

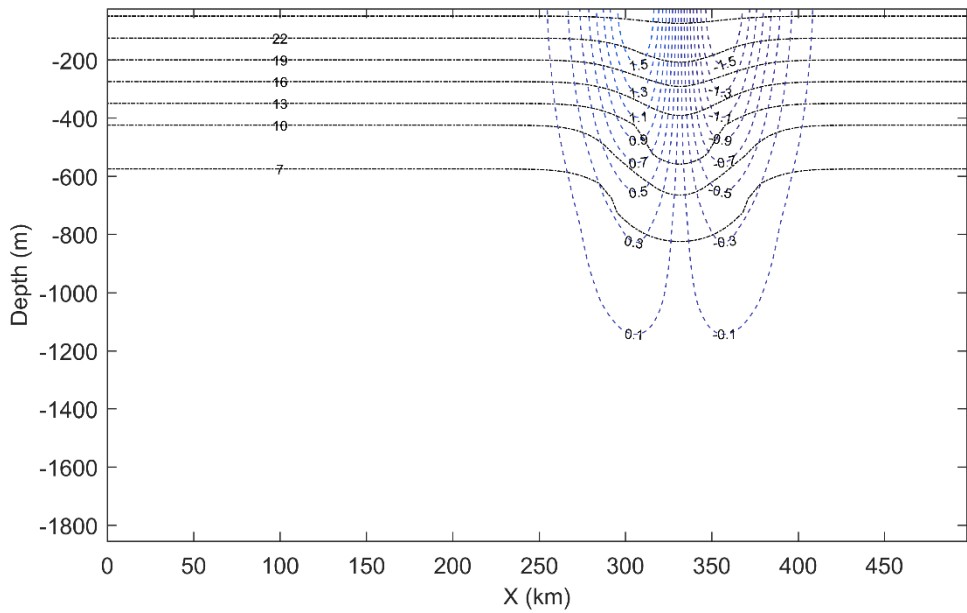

**Figure 4. Initial velocity (m/s) and temperature (℃) profiles of the warm eddy used in the model.**



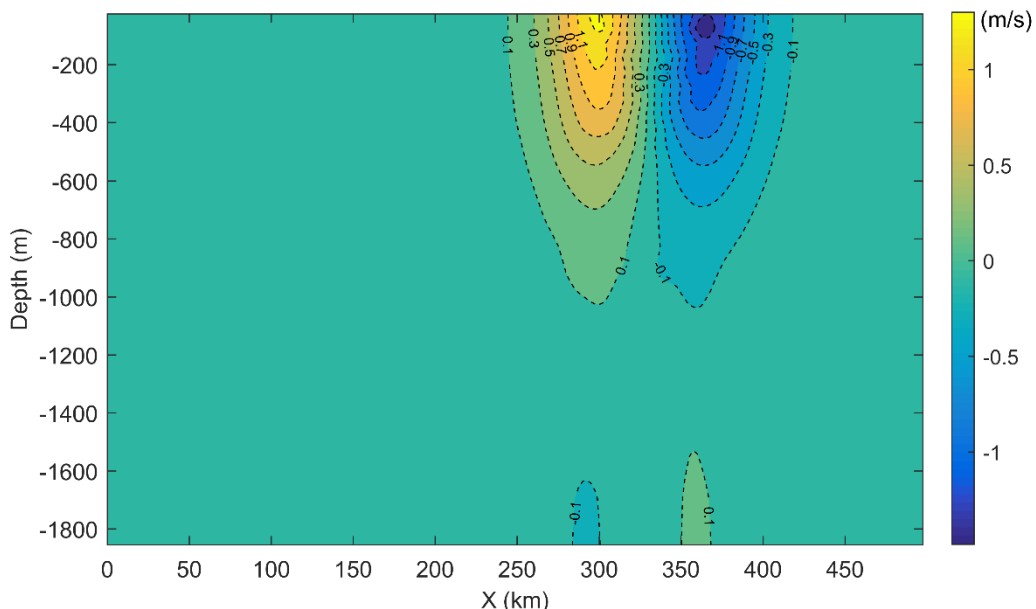

**Figure 5. The transect of velocity distribution through the eddy centre (on the f plane).**

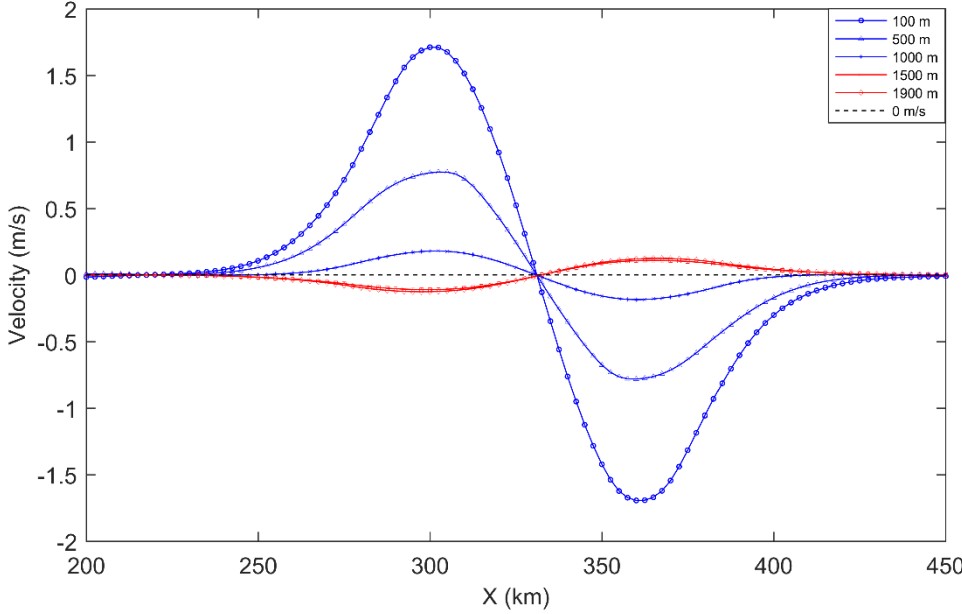

**Figure 6. The V-velocity distribution along x-axis through the eddy centre at different depths (on the f plane).**



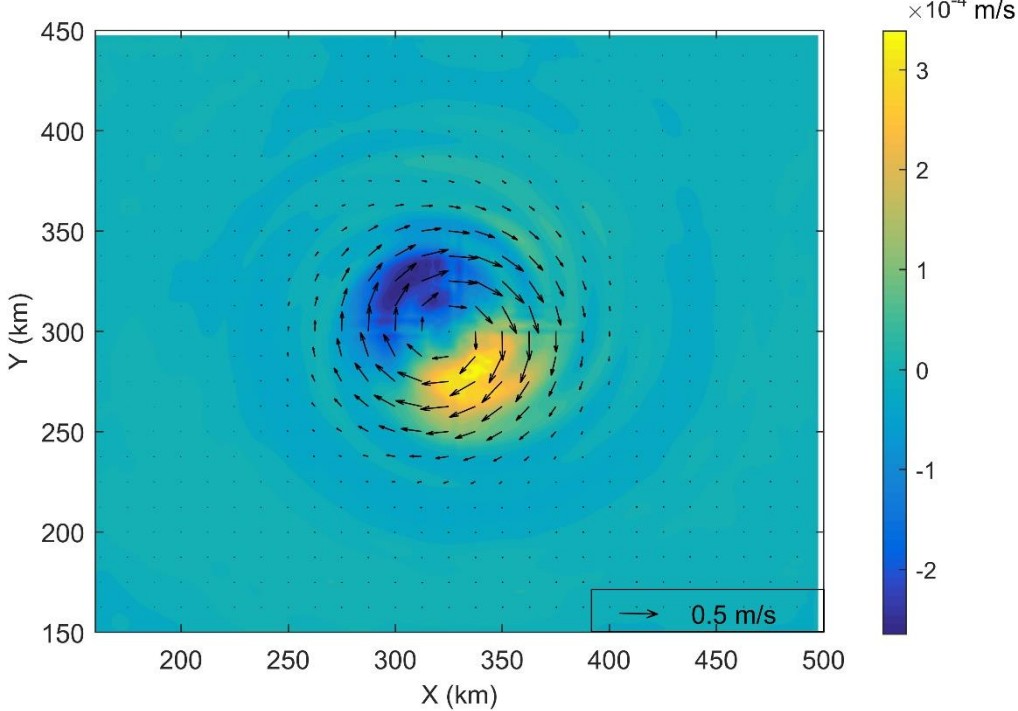

**Figure 7. The velocity distribution of the eddy at 500 m depth. Black arrows, whose length is proportional to velocity, denote the current direction. The colours represent the vertical velocity distribution (on the β plane).**

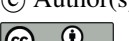


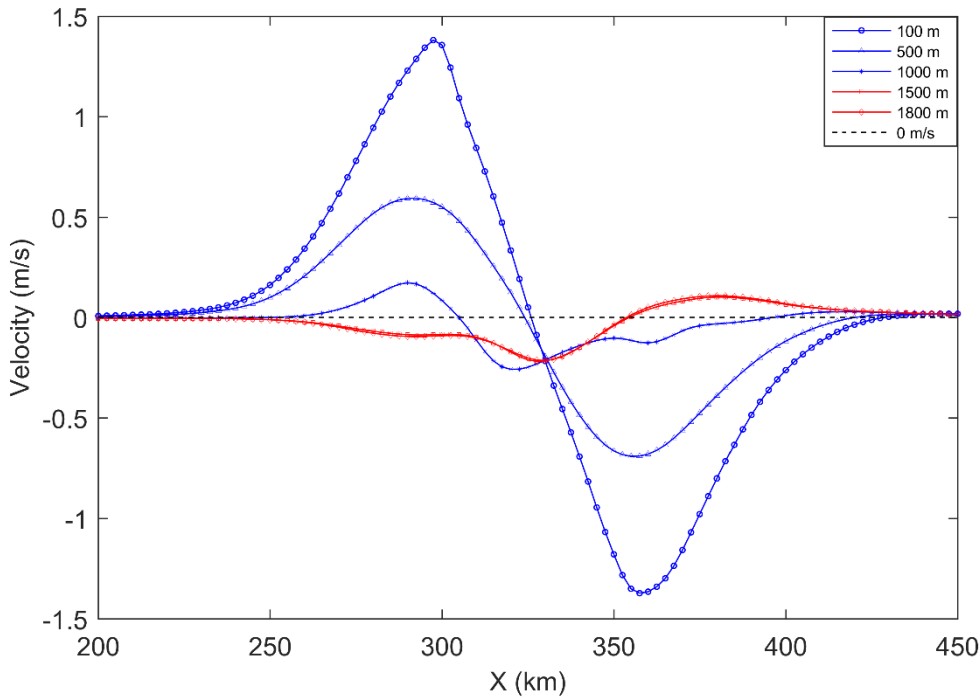

**Figure 8. The v-velocity distribution along x-axis through the eddy centre at different depths (on the β plane).**

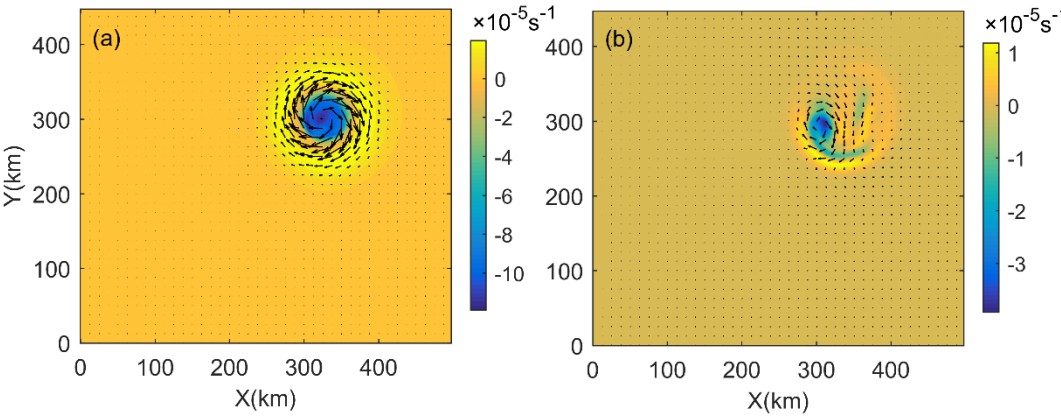

**Figure 9. The relative vorticity distribution of the eddy at (a) 100m and (b) 1000m depth (on the β plane).**



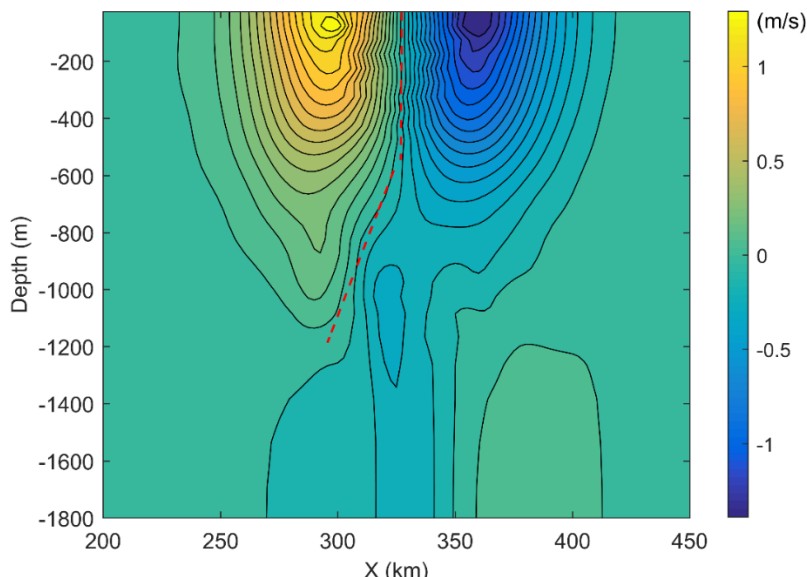

**Figure 10.East-west transect of v-velocity distribution through the eddy centre. The red dot line indicates the position of the eddy axis.**

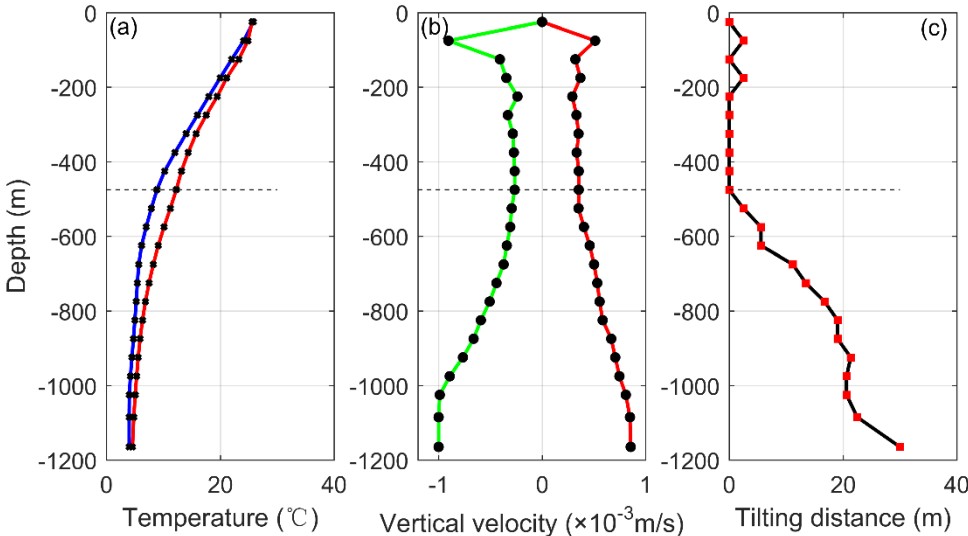

5    **Figure 11. (a) The temperature profile (red line represents temperature at the eddy centre and blue line represents temperature distribution of background field). (b) The maximum vertical velocity profile (red line represents upward velocity and green line represents downward velocity). (c) The tilting distance to west along the depth. The dot line is 475 m water depth.**


**Figure 12. The v-velocity distribution along x-axis through the eddy centre at different depths with a linear stratification (on the β plane).**

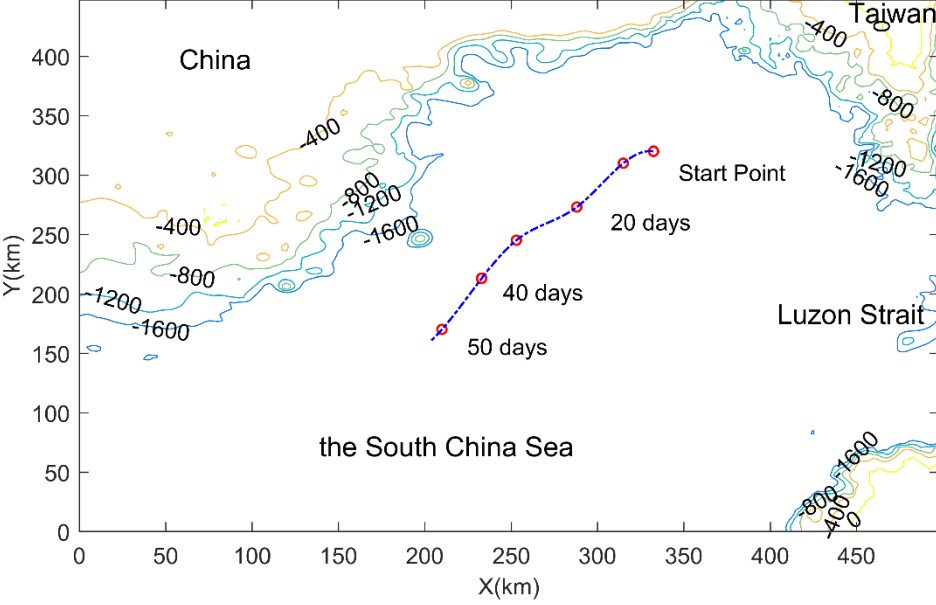

**Figure 13. Eddy trajectory in the northern South China Sea. The eddy centre is depicted by circles every 10 days. Water depths are in metres.**





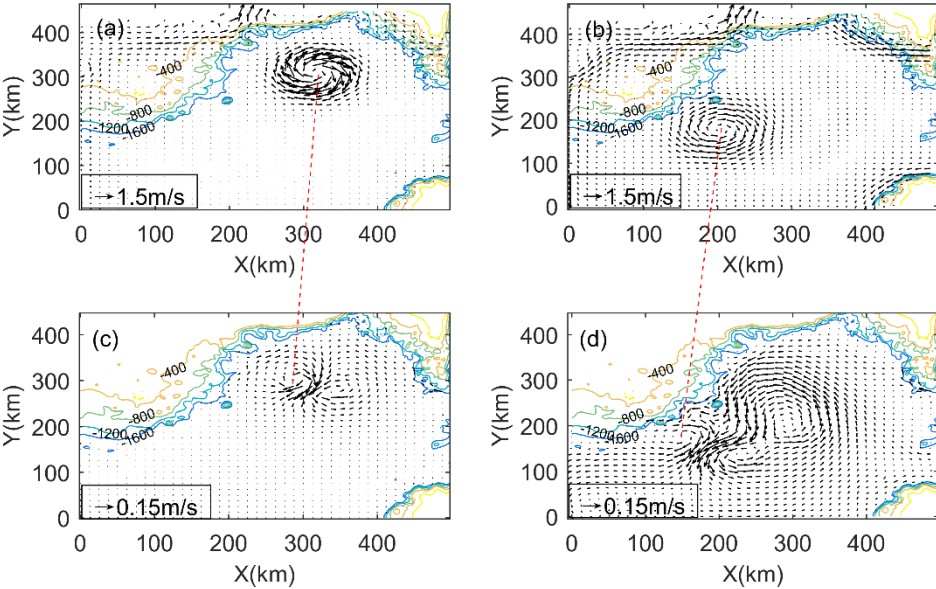

**Figure 14. The eddy structure in the northern South China Sea. Left panels are the velocity field at day 10 at a depth of (a) 100m and (c) 1000m. Right panels are the velocity field at day 50 at a depth of (b) 100m and (d) 1000m. The red dot line relates the positions of the eddy centres at depths of 100 m and 1000 m. Isobaths depths are in metres.**

