# Peer review of "The impact of the planetary $\beta$ -effect on the tilting vertical structure of a mesoscale eddy"

_Ocean Science, 2018_

## Referee Comment (RC1) · Anonymous Referee #1 · 2 Jul 2018

The concept explored by the authors is interesting, and worthy of eventual publication. Despite my enthusiasm for the topic, I have some substantial concerns that I believe must be addressed prior to publication.

**1  Substantial issues:**

1) The authors present several simulations as evidence to support their assertion that beta affects the tilting of the vertical axis of mesoscale eddies, and indeed these simulations do show differences when beta is included or excluded. However, the impact is not quantified, either through analytical scaling arguments (which would greatly improve section 2) or a numerical sweep of the parameter space. I would ideally like to see the inclusion of both a scaling argument and the results from multiple simulations in which beta is varied in a systematic way and its effect is quantised.

2) The impact of stratification is mentioned several times in the manuscript, and yet it is not quantified anywhere. Including a thermocline appears to suppress the tilting of the axis in the upper ocean (figures 10 and 11 show that most of the tilt appears below the thermocline). The simulation with a linear stratification is not described in much detail, but it seems as though the linear stratification reduced the tilt of the eddy axis (10 km tilt compared with 35 km when a thermocline was present). I would once again like to see some rigorous analysis and/or a series of simulations in which the stratification is systematically varied. The authors address this very briefly in final sentence of section 4 "… it is evident that the ocean stratification is an important factor in controlling the vertical structure of a mesoscale eddy" but I would like to see a more rigorous analysis.

3) Interactions with the bathymetry are mentioned several times in the manuscript, but there is no evidence that the authors have attempted to systematically address the influence of the sea floor on the tilting of the vertical axis. They show that using a realistic bathymetry map alters the tilt of the axis, but there is no indication as to why. Is it the slope perpendicular to the direction of travel? Is it because the presence of a slope forces the eddy to cross lines of latitude? An ensemble of simulations in which the direction and steepness of the slope are varied would go a long way towards answering these questions.

4) The description of their numerical setup is incomplete. Someone reading this manuscript has almost no chance of reproducing the simulations. For example, what form does a(z) from equation 4 take? What equation of state does the model use? Which advection scheme do they use? These are important numerical decisions that can substantially affect the results of numerical experiments, and the authors do not provide enough details for their readers to assess the results.

To summarise these concerns: the topic of the manuscript is very interesting, but the analysis presented lacks sufficient rigour to be published in its current form.

**2 Minor concerns:**

1) I found the paragraph encompassing lines 13-20 on page 3 to be extremely vague. As mentioned above section 2 would be greatly improved by the inclusion of a rigorous scaling argument showing the expected impact of beta, stratification, and bathymetry on the tilting of the vertical axis.

2) Page 7 lines 17-18: Why are the authors correlating vertical velocity and axis displacement? This statistical test lacks context. Furthermore, the authors present a correlation between vertical velocity and axis tilt as evidence to support their argument, and yet examining figure 11 clearly shows that the vertical velocity is non-zero at depths where the axis is still vertical. Given their argument that non-zero vertical velocities cause the axis tilting, this discrepancy should be discussed.

3) I found many of the figures difficult to interpret. I would appreciate longer, more descriptive captions.

4) Figure 1 is meant to show the asymmetry in the pressure field and the vertical motion. The asymmetry is adequately conveyed, but there is no way for the reader to learn anything about the vertical motion from what is plotted.

5) Figures 5, 7, 9, and 10 would benefit from a divergent colourmap – zero is an important value in these figures, and it would be good to make it clear.

6) Figure 13 – the authors describe the eddy's propagation as following a contour of ocean depth, yet the figure does not show any contours in the region of the eddy. The contour interval should be adjusted to show whether the eddy does indeed follow contours of the ocean floor.

7) Minor numerical issues:

a. MITgcm is a z-coordinate model. It has levels, not layers. The references to model layers within the manuscript should be changed to refer to levels.

b. The vertical resolution of the model is quite coarse, especially in comparison to the horizontal resolution (see Stewart et al. (2017) for a discussion of vertical resolution in ocean models). The authors should either use a more appropriate vertical resolution, or justify their choice to use only 28 levels with such a fine horizontal resolution.

c. The authors should cite at least one of the papers describing the development of MITgcm, e.g. Marshall et al. (1997).

d. Using a bottom-drag of some sort would likely damp away the bottom intensified counter-rotating eddies.

8) A brief description of why anticyclones are much more prevalent in the South China Sea would allow a wider range of readers to engage with the manuscript.

9) The simulation on an f plane is not particularly informative. It should be an end member of a series of simulations in which beta is systematically varied, not an entire section.

10) Page 6 lines 16-18: what happens to the vertical velocity at the bottom? The text indicates that it increases with depth, but given the flat bottom it must be zero at the bottom.

11) Page 7 line 1: what baroclinic instability are the authors referring to here?

**3 References:**

Marshall, J. C., Adcroft, A., Hill, C., Perelman, L., Heisey, C. (1997). A finite-volume, incompressible Navier Stokes model for studies of the ocean on parallel computers. Journal of Geophysical Research: Oceans, 102(C3), 5753–5766. http://doi.org/10.1029/96JC02775

Stewart, K. D., Hogg, A. M., Griffies, S. M., Heerdegen, A. P., Ward, M. L., Spence, P., England, M. H. (2017). Vertical resolution of baroclinic modes in global ocean models. Ocean Modelling, 113, 50–65. http://doi.org/10.1016/j.ocemod.2017.03.012

---

## Referee Comment (RC2) · Anonymous Referee #2 · 20 Jul 2018

The paper studies how the planetry beta-effect influences the vertical structure of mesoscale eddies. The work is original, and thus worth publication. However, I have some issues which need addressing before publication:

1. Minor: The authors should define the scale of mesoscale eddies. 2. Major: The authors should compare MITgcm simulations with the observations of tilted mesoscale eddies in the South China Sea, otherwise it is difficult to judge the results.

---

## Author Comment (AC1) · 1 Sep 2018

We are glad to read the referee's comments, such as 'the concept explored by the authors is interesting and worthy of eventual publication'.

Substantial issues: 1) The authors present several simulations as evidence to support their assertion that beta affects the tilting of the vertical axis of mesoscale eddies, and indeed these simulations do show differences when beta is included or excluded. However, the impact is not quantified, either through analytical scaling arguments (which would greatly improve section 2) or a numerical sweep of the parameter space. I would ideally like to see the inclusion of both a scaling argument and the results from multiple simulations in which beta is varied in a systematic way and its effect is quantized.

[Figure]

Reply: Motivated by the observation of the eddy's tilting axis in the South China Sea (Zhang et al., 2016), the influence of planetary $\beta$ on eddy vertical structure evolution is studied based on a simple theoretical analysis and numerical model experiments. Because it's difficult to obtain an analytic solution of eddy tilt under the influence of $\beta$ effect, our prime purpose of this study is to describe the $\beta$-effect on vertical structure of mesoscale eddy qualitatively. According to your suggestion, we will expand the numerical model experiments with the value of $\beta$ varying in a systematic way to quantify its effect on the tilt of a mesoscale eddy.

2) The impact of stratification is mentioned several times in the manuscript, and yet it is not quantified anywhere. Including a thermocline appears to suppress the tilting of the axis in the upper ocean (figures 10 and 11 show that most of the tilt appears below the thermocline). The simulation with a linear stratification is not described in much detail, but it seems as though the linear stratification reduced the tilt of the eddy axis (10 km tilt compared with 35 km when a thermocline was present). I would once again like to see some rigorous analysis and/or a series of simulations in which the stratification is systematically varied. The authors address this very briefly in final sentence of section 4 "…it is evident that the ocean stratification is an important factor in controlling the vertical structure of a mesoscale eddy" but I would like to see a more rigorous analysis.

Reply: In the study of eddy tilt caused due to the $\beta$-effect, model results show that the axis tilt mainly occurs below the thermocline. In order to show ocean stratification do have an impact on vertical structure tilt of a mesoscale eddy, for comparison, a linear vertical temperature profile without a strong thermocline is applied in the numerical model. Apologize for not including a clear description in the simulation with the linear stratification in this paper. As your suggestion, we will carry out detailed analysis of the effects of the stratification on eddy tilt through a series of experiments. The model results will be presented in the manuscript in detail.

3) Interactions with the bathymetry are mentioned several times in the manuscript, but there is no evidence that the authors have attempted to systematically address the

influence of the sea floor on the tilting of the vertical axis. They show that using a realistic bathymetry map alters the tilt of the axis, but there is no indication as to why. Is it the slope perpendicular to the direction of travel? Is it because the presence of a slope forces the eddy to cross lines of latitude? An ensemble of simulations in which the direction and steepness of the slope are varied would go a long way towards answering these questions.

Reply: According to the analyzation on dynamics of mesoscale eddy, many factors would influence the structure of a mesoscale eddy in oceans and topography is one of them. In addition to the topographic $\beta$ effect, the nonlinear interaction between a mesoscale eddy and topography is a more complicated process which could impact the structure of the eddy. In this article, our main attention will focus on the influence of planetary $\beta$ on eddy vertical structure evolution, and the real topography of the South China Sea is used to show the consistency of the numerical results with the observed results on the $\beta$ plane, though there are differences in tilt distance. The influence of topography on vertical structure of a mesoscale eddy involves many aspects, as you said, including directions and steepness of the slope. Therefore the effect of bottom topography on the vertical structure of an eddy merits further investigation and this is an aspect in our next phase of the work.

4) The description of their numerical setup is incomplete. Someone reading this manuscript has almost no chance of reproducing the simulations. For example, what form does a(z) from equation 4 take? What equation of state does the model use? Which advection scheme do they use? These are important numerical decisions that can substantially affect the results of numerical experiments, and the authors do not provide enough details for their readers to assess the results.

Reply: The description of the numerical setup will be improved in reversion. Key parameters and main equations will be presented in the article. Thank you for pointing out our deficiencies in the article.

2 Minor concerns: 1) I found the paragraph encompassing lines 13-20 on page 3 to be extremely vague. As mentioned above section 2 would be greatly improved by the inclusion of a rigorous scaling argument showing the expected impact of beta, stratification, and bathymetry on the tilting of the vertical axis.

Reply: The part of the text which is not clear will be improved. We will perform a series of new experiments to analyze the effects of planetary $\beta$ and ocean stratification on eddy tilt systematically. The model results will be presented in the reversion.

2) Page 7 lines 17-18: Why are the authors correlating vertical velocity and axis displacement? This statistical test lacks context. Furthermore, the authors present a correlation between vertical velocity and axis tilt as evidence to support their argument, and yet examining figure 11 clearly shows that the vertical velocity is non-zero at depths where the axis is still vertical. Given their argument that non-zero vertical velocities cause the axis tilting, this discrepancy should be discussed.

Reply: According to the analysis based on a simple theory, the tilt of the vertical axis of a mesoscale eddy is triggered by vertical motion which is caused by the planetary $\beta$ effect and nonlinear dynamics of mesoscale eddy. The model results show that the maximum vertical velocity and the axis displacement have similar variation with depth, therefore, a correlation between vertical velocity and axis tilt was present. Actually, the pressure anomaly induced by vertical velocity gradient is the direct effect of eddy tilt, therefore, correlation analysis between the gradients of vertical velocity and tilting distance with depth will be more consistent with the mechanisms of eddy tilt.

3) I found many of the figures difficult to interpret. I would appreciate longer, more descriptive captions.

Reply: Captions in the article will be re-written.

4) Figure 1 is meant to show the asymmetry in the pressure field and the vertical motion. The asymmetry is adequately conveyed, but there is no way for the reader to

learn anything about the vertical motion from what is plotted.

Reply: Figure 1 and related expressions in the text will be modified.

5) Figures 5, 7, 9, and 10 would benefit from a divergent colourmap – zero is an important value in these figures, and it would be good to make it clear.

Reply: Thanks for the suggestions. We will mark the distribution of zero value in these figures.

6) Figure 13 – the authors describe the eddy's propagation as following a contour of ocean depth, yet the figure does not show any contours in the region of the eddy. The contour interval should be adjusted to show whether the eddy does indeed follow contours of the ocean floor.

Reply: Figure 13 will be re-drawn to show contours of ocean depth clearly in all regions.

7) Minor numerical issues: a. MITgcm is a z-coordinate model. It has levels, not layers. The references to model layers within the manuscript should be changed to refer to levels. b. The vertical resolution of the model is quite coarse, especially in comparison to the horizontal resolution (see Stewart et al. (2017) for a discussion of vertical resolution in ocean models). The authors should either use a more appropriate vertical resolution, or justify their choice to use only 28 levels with such a fine horizontal resolution. c. The authors should cite at least one of the papers describing the development of MITgcm, e.g. Marshall et al. (1997). d. Using a bottom-drag of some sort would likely damp away the bottom intensified counter-rotating eddies.

Reply: a. The related terms will be modified. b. Yes, for the consideration of calculation capacity, 28 uneven levels used in vertical dimension in the model is a bit coarse, but it basically satisfies the needs of the current research. c. Thank you for your suggestion, this part of the content will be revised in the revision. d. Yes. The bottom counter-rotation eddy is caused by the adjustment of geostrophic balance in model initialization. This kind of eddy won't affect our conclusions on the evolution of its vertical structure.

8) A brief description of why anticyclones are much more prevalent in the South China Sea would allow a wider range of readers to engage with the manuscript.

Reply: Thank you for the suggestion, we will provide a supplement on anticyclones in the South China Sea.

9) The simulation on an f plane is not particularly informative. It should be an end member of a series of simulations in which beta is systematically varied, not an entire section.

Reply: We will systematically analyze the influences of $\beta$-effect on eddy tilt through a series of experiments with varying $\beta$. The article will be modified to include these new results.

10) Page 6 lines 16-18: what happens to the vertical velocity at the bottom? The text indicates that it increases with depth, but given the flat bottom it must be zero at the bottom.

Reply: Vertical velocity increases with depth in the area of above 1200 m, decreases below 1400 m and is zero at bottom. We will add a statement on this in the article.

11) Page 7 line 1: what baroclinic instability are the authors referring to here?

Reply: The phrase 'baroclinic instability' here may not be accurate, so we will remove the sentence in the revision.

---

## Author Comment (AC2) · 1 Sep 2018

We are glad to read the referee's comments such as 'the work is original, and thus worth publication'.

1. Minor: The authors should define the scale of mesoscale eddies.

Reply: We will illustrate the scale of the mesoscale eddies in the reversion.

2. Major: The authors should compare MITgcm simulations with the observations of tilted mesoscale eddies in the South China Sea, otherwise it is difficult to judge the results.

Reply: Thank you for the suggestion, we will include the comparison between the

simulation results and the observations of tilted mesoscale eddies in the South China Sea in the revision.

Once again, thank-you very much for the two referees' constructive comments and suggestions.
* * *